# A prospective, multicenter study of invasive fungal disease caused by molds in children and adults in Chile

Ricardo Rabagliati[1,2,3]*, Dona Benadof[3,4,5], Luis Thompson[3,6], Patricia Garcia[2,3,7], Tamara Gonzalez[2,3,7], Cecilia Vizcaya[2,3,8], Andres Soto[3,9], Catalina Gutierrez[3,10], Lorena Rodriguez[3,10], Yenis Labraña[3,10], Cecilia Tapia[3,11], Marcela Zubieta[3,12], Martin Lasso[1,3,13], Valentina Gutierrez[3,8,13], Veronica Contardo[3,14], Yocelyn Castillo[3,15], Karen Ducasse[3,16], Mario Calvo[3,17], Romina Valenzuela[3,18], Maria Elena Santolaya[3,18], on behalf of the Chilean Invasive Mycosis Network[¶]

1 Department of Infectious Diseases, Faculty of Medicine, Pontificia Universidad Católica de Chile, Santiago, Chile, 2 Red de Salud UC CHRISTUS, Santiago, Chile, 3 Chilean Invasive Mycosis Network, Santiago, Chile, 4 Microbiology Laboratory, Hospital Dr. Roberto del Río, Santiago, Chile, 5 Department of Pediatric and Surgery, Faculty of Medicine, Universidad de Chile, Santiago, Chile, 6 Infectious Diseases Unit, Department of Medicine, Clínica Alemana, Universidad del Desarrollo, Santiago, Chile, 7 Department of Clinical Laboratories, Faculty of Medicine, Pontificia Universidad Católica de Chile, Santiago, Chile, 8 Department of Pediatric Infectious Diseases and Immunology, Faculty of Medicine, Pontificia Universidad Católica de Chile, Santiago, Chile, 9 Department of Internal Medicine, Campus Oriente, Faculty of Medicine, Universidad de Chile. Service of Medicine, Hospital del Salvador, Santiago, Chile, 10 Department of Medicine, Hospital San Juan de Dios, Santiago, Chile, 11 Microbiology Laboratory, Clínica Dávila, Santiago, Chile, 12 Department of Pediatrics, Hospital Dr. Exequiel González Cortés, Santiago, Chile, 13 Pediatric Infectious Diseases Unit, Hospital Dr. Sótero del Río, Santiago, Chile, 14 Department of Pediatrics, Hospital Dr. Roberto del Río, Santiago, Chile, 15 Hospital de Antofagasta, Antofagasta, Chile, 16 Department of Pediatrics. Hospital Gustavo Fricke. Viña del Mar, Chile, 17 Institute of Medicine, Universidad Austral de Chile. Hospital de Valdivia, Valdivia, Chile, 18 Department of Pediatrics, Research Unit, Hospital Dr. Luis Calvo Mackenna, Faculty of Medicine, Universidad de Chile, Santiago, Chile

¶ The complete membership of the author group can be found in the Acknowledgments section.
* rabagli@ucchristus.cl

## Abstract

### Background

Invasive mold diseases (IMDs) are a severe complication of immunocompromised subjects and an emerging problem among severely ill, apparently immunocompetent patients. The aim of this study was to describe the epidemiological and clinical features of IMDs in Chile.

### Methods

Prospective study of IMD cases in children and adults from 11 reference hospitals in Chile from May 2019 to May 2021.

### Results

One hundred seventy-six cases were included, 135 in adults and 41 in children, with an overall incidence of 0.4/1,000 admissions. The median age was 10.5 years in

**Data availability statement:** All relevant data are within the paper and its Supporting Information files.

**Funding:** This study was partially supported by FONDECYT, grant number 1200964, and partially supported by an independent medical grant to the Chilean Invasive Mycosis Network from Pfizer Chile, BioMerieux Chile, and Gador. The funders had no role in the study design, data collection, analysis, and manuscript preparation We have included the Gador Letter as "other document" and we explained in the cover letter that we do not have a funding letter for the Chilean Invasive Mycosis Network from Pfizer Chile and BioMerieux Chile.

**Competing interests:** The authors would like to state the following competing interests: Chilean Invasive Mycosis Network received funding from Pfizer Chile, BioMérieux Chile, and Gador. There are no patents, products in development, or marketed products associated with this research to declare. The funders had no role in study design, data collection and analysis, decision to publish, or preparation of the manuscript.

children and 56.6 years in adults, with male gender predominance in adults (61.5% versus 41.5%, p = 0.03). Immunosuppression was the most common condition in both children and adults. However, cancer, neutropenia, and hematopoietic cell transplantation were significantly more frequent in the pediatric group. In contrast, diabetes, viral pneumonia, chronic kidney disease, and chronic obstructive pulmonary disease were significantly more frequent among adult patients. Regarding the diagnostic category, 30.1% of cases were proven, 55.7% probable and 14.2% possible. Aspergillosis was the most frequent IMD diagnosed in 75.5% of cases, followed by fusariosis in children and mucormycosis in adults. Viral pneumonia was associated in 40.3% of cases, mainly COVID-19, with aspergillosis in 87.3%. No triazole resistance was observed in *Aspergillus* spp.. Antifungals were prescribed in 97.2% of the patients: voriconazole 61.4%, liposomal amphotericin 20.5%, combination antifungals 11.1%, and others 6.4%. Overall survival was 68.7%, 61.4%, and 51.7% at 30, 90 and 180 days, respectively.

## Discussion

This is the most extensive study of IMDs in Chile, evidencing an incidence of 0.4 per 1,000 admissions, with aspergillosis being the most frequent infection. Nearly 40% of cases were associated with respiratory viruses, accounting for the impact of COVID-19. Despite almost all patients starting antifungal therapy, the survival rate was poor. It is advisable to start a surveillance program of IMDs in Chile and verify the absence of azole resistance of *Aspergillus* spp.

## Introduction

Invasive mold diseases (IMDs) are a severe infectious complication in immunocompromised patients [1]. In fact, most filamentous fungi behave as opportunist agents and a relevant cause of infection in individuals with hematological cancer, hematopoietic cell transplantation (HCT), solid organ transplant (SOT) recipients, and steroid or immunosuppressor users [2–7]. Nevertheless, in recent years, an increasing number of cases have been described in apparently immunocompetent people, particularly severely ill patients in intensive care units affected by diabetes, chronic obstructive pulmonary disease (COPD), hepatic failure, or chronic kidney disease (CKD) [8–12], and associated with severe viral respiratory infections as influenza [13,14], and very recently cases described in patients with severe pneumonia due to SARS-CoV-2 during the COVID-19 pandemic [15–17].

Usually, IMDs are not part of the infection surveillance systems worldwide; therefore, the epidemiological information originates from case reports in specific populations, with biases in patient selection and diagnostic capacities. Recently, through the initiative of the Leading International Fungal Education (LIFE), the burden of fungal infection at the population level of different countries in Africa, America, Asia, and Europe has been published, including estimates for yeast and mold infections

[18–22]. In Chile, the incidence was estimated at 1.7 cases of aspergillosis per 100,000 inhabitants and 0.2 cases of mucormycosis per 100,000 inhabitants per year [23]. On the other hand, based on documented data in hematological patients in Chile, it has been previously reported that up to 5.8% and 20% of the invasive fungal diseases occur in children and adults with febrile neutropenia (FN), respectively, being aspergillosis the most prevalent IMD [24,25]. However, national epidemiological data is unknown. Additionally, we do not have information on the emerging problem of *Aspergillus* resistance to azoles in our country, compared with some data published from other Latin American countries confirming the existence of *A. fumigatus* resistance to voriconazole in the region [26–28].

The Chilean Invasive Mycosis Network was created in 2011 to implement active surveillance of different mycoses in children and adults nationwide. In 2019, we described the epidemiology of candidemia [29], and after that, we defined IMDs epidemiology research as a new challenge.

The main objective of this study was to describe the epidemiological and clinical features of IMDs cases in tertiary-level hospitals in Chile, comparing children and adults, trying to identify special issue in each group. The relevance of making these data is to raise awareness of these infections, to add epidemiological information from IMDs in Latin America, to stress the importance of fungal infection surveillance, to improve the diagnostic tools availability, to search for the existence of triazoles resistance, to improve access to antifungal therapies, to generate the essential information to design national management guidelines and define policies for the prevention of fungal infections in the different care centers.

## Patients and methods

### Overall study design

A prospective, multicenter, observational study of IMDs cases in adults and children was conducted from May 2019 to May 2021 in 11 public and private hospitals at the tertiary level of care in Chile, belonging to the Chilean Invasive Mycosis Network, of which eight are located in Santiago: UC-CHRISTUS (UC), Salvador (S), San Juan de Dios (SJD), Dr. Sótero del Río (SR), Dr. Luis Calvo Mackenna (LCM), Dr. Exequiel Gonzáles Cortes (EGC), Dr. Roberto del Río (RR), Dávila (D); and three hospitals are located out of Santiago: Antofagasta (A) from the North, Gustavo Fricke (GF) from the Center and Valdivia (V) from the South of Chile. Nine centers are public hospitals, 2 (UC, D) are from the private health system, 3 (LCM, EGC, and RR) are pediatric centers, and the others are adult and pediatric hospitals. Before beginning data collection, all local Institutional Review Boards (IRB) from the participating healthcare centers approved the protocol and waived consent for enrollment.

At the beginning of the study, a survey was conducted to determine the more frequent patient conditions in each hospital, the microbiological diagnostic capabilities, the availability of images and antifungal therapies, and the antifungal prophylaxis policies.

Each center's local research team, consisting of a clinician and a microbiologist, identified the cases. The following data were collected in an ad hoc case report form: age, gender, previous use of antibacterial and antifungal drugs; baseline conditions, including cancer, leukemia/lymphoma, neutropenia, HCT, SOT, viral pneumonia, COPD, diabetes, CKD, previous use of antifungal, corticosteroids and immunosuppressors drugs, HIV infection; clinical presentation imaging and microbiological results at the time of IMDs diagnosis; treatment, including antifungal therapy and voriconazole plasmatic levels. For outcomes, we evaluated hospital length of stay (LOS), the complete response at 6 and 12 weeks of antifungal therapy, and 30, 90, and 180-day survival rates after IMD diagnosis. The IMDs cases and their diagnostic certainty were reviewed by two investigators (RR and MS).

### Laboratory procedures

The microbiological diagnostic approach was conducted at each healthcare center, including direct sample staining and fungal culture from respiratory samples, secretions, tissues, or other samples. For tissues, a sterile tube was used to take

samples, which were cut into pieces to be sown on plates or tubes of sabouraud dextrose 4% agar. The identification of filamentous fungi was carried out using conventional methods, such as macroscopy and microscopy. With respect to the fungi cultures, they were incubated at 30 °C in plates containing Sabouraud dextrose agar at 2% and potato dextrose agar (Merck[R], Germany). Incubation was maintained for up to 30 days, with daily macroscopic evaluations to monitor fungal growth and morphological characteristics. Microscopic observation was done using commercial lactophenol cotton blue stain (Merck[R], Germany), applied between a slide and coverslip, and examined under light microscopy at 40X magnification. Additionally, MALDI-TOF mass spectrometry (Bruker Microflex LT with Fungi Library 3.0) was used for final identification using the tube protein extraction method. Protein extraction was performed following the Bruker protocol for filamentous fungi, with minor modifications. Biomass obtained from a 48-hour subculture in Sabouraud broth at 30 °C was harvested by centrifugation and washed twice with 1000 µL of 70% ethanol. The resulting pellet was resuspended in 10–50 µL of 70% formic acid and mixed by vigorous agitation, followed by the addition of an equal volume of acetonitrile and a second round of mixing. After final centrifugation, 1 µL of the supernatant was applied to a steel target plate, and once dry, 1 µL of α-cyano-4-hydroxycinnamic acid (HCCA) matrix was added for MALDI-TOF MS analysis.

For the RT-PCR of *Aspergillus*, we used the *Aspergillus* spp ELITe MGB® Kit, a PCR assay for the detection and quantification of DNA from the genus *Aspergillus*, including the species: *A. niger*, *A. nidulans*, *A. terreus*, *A. flavus*, *A. versicolor*, and *A.glaucus*.

*Aspergillus* spp. isolates were sent to the reference laboratory (UC) for antifungal susceptibility tests. If the test was unavailable, then the isolates were preserved in an Eppendorf tube with sterile water stored at 4°C. For applying for the susceptibility test, the isolated were plated on both potato dextrose agar and sabouraud dextrose 4% agar. The susceptibility test was carried out using the commercial panel AST YO9 Sensititre™ YeastOne™ Plate, according to the manufacturer's recommendations to obtain the minimal inhibitor concentration (MIC). The epidemiological cut-off value (ECV) for each *Aspergillus* species was obtained through the CLSI standard M57SEd4E [30].

Serum or bronchoalveolar lavage (BAL) galactomannan (GM) was conducted at each center. However, two laboratories (UC and LCM) were available for additional testing if required for treating team, including *Aspergillus*-PCR, fungal identification in challenging cases, voriconazole plasma levels or 1–3 β-D-glucan (BDG) (UC).

## Definitions

*Children*: Patients < 18 years of age. *Taxonomy*: We used the traditional names for fungal identification, including genus, family, section, and species [31,32]. *IMD cases:* Were defined based on the presence of host factors, clinical presentation, and radiological findings, in combination with microbiological evidence (e.g., direct microscopy, culture, fungal biomarkers) and/or histopathological confirmation. We applied the most recent definitions proposed by the European Organization for Research and Treatment of Cancer/Mycoses Study Group (EORTC/MSG) [33], classifying cases as follows:

- **Proven IMD**: Histopathological evidence of fungal invasion, a positive culture from blood or a normally sterile site, or detection of fungal DNA by PCR in tissue.

- **Probable IMD**: Presence of host factors, compatible clinical and imaging findings, and mycological evidence such as fungal biomarkers, culture, or PCR.

- **Possible IMD**: Presence of host factors plus clinical features or imaging findings suggestive of fungal infection, but without mycological evidence.

In a similar manner, the consensus definitions were applied to cases of COVID-19-associated pulmonary aspergillosis (CAPA) [34]. *Mold coinfection*: In cases where more than one mold was identified, we considered the local researcher's opinion to confirm the coinfection. *Breakthrough invasive mold diseases (b-IMDs)*: IMDs cases diagnosed in patients receiving systemic antifungal therapy for at least 7 days before the diagnosis, independent of the antifungal prescription

reason [35]. *Complete response*: The resolution of clinical manifestation and radiological abnormalities attributable to IMDs and, if available, evidence of mycological clearance [36].

### Statistical analysis

Patient demographics and characteristics of the infection episodes were presented using counts and percentages for categorical data and means ± standard deviations (SD) or medians and interquartile ranges (IQR) for continuous data. The chi-squared or Fisher's exact test was used for dichotomous variables and the t-test or U de Mann-Whitney for continuous variables. Statistical analysis was performed using Stata SE 14.0 software for Mac (College Station, TX77845, USA). A *p*-value ≤0.05 was considered statistically significant.

## Results

Two hundred and two IMD cases were reported, of which 26 were excluded as they were considered contaminations; therefore, 176 cases, 135 in adults and 41 in children, were included. These cases were detected among 437,447 hospital admissions in the study period, with an overall incidence of 0.4/1,000 admissions.

### General characteristics

Table 1 shows the general characteristics of the episodes of IMDs and compares them to those of children and adults. The median age was 11 and 56.5 years old in children and adults, respectively, with male gender predominated in adults (61.5% versus 41.5%, p = 0.03). IMDs cases were predominant among patients with hematological malignancies (acute lymphocytic leukemia in 32, acute myeloid leukemia in 21, lymphoma in 8 subjects) over cases with solid organ tumors (4 patients). The IMDs cases in HCT were presented with an average of 198 days post-transplant, and 6 cases suffered graft versus host disease. Eight cases were in patients with SOT: six cases of kidney, one pancreatic kidney, and one heart transplantation, all of them in adults.

Notably, there were statistically significant differences between children and adults. Immunosuppression-related conditions, like cancer, neutropenia, and HCT, occurred more frequently in the pediatric group than in the adult group. On the contrary, diseases such as diabetes, CKD, and COPD occurred more frequently among adult patients. Among diabetic patients, 22 (44%) had glycosylated hemoglobin over 7%. Between 15 patients with CKD, 8 (53.3%) received dialysis, and the 5 HIV-infected adults had a median CD4 T lymphocyte count of 7.2 (4–39) cells/mm$^3$.

There were 81 (46%) cases hospitalized in ICU at the time of diagnosis of IMD (29.3% of cases in children and 51.1% of adults p = 0.02).

### Clinical features

In all the 176 IMD cases, a focus on infection was identified. In 145 (82.4%), it was limited to one organ, and in 31 (17.6%), more than one site of infection was identified. Among those limited to one organ, the more frequent focus was the lung in 113 (64.2%), followed by sinus in 18 (10.2%), and skin and soft tissue in 7 (4%). In cases with more than one focus, 22 (12.5%) corresponded to pulmonary involvement plus another single or multiple additional focuses (sinus in 13; hepatic in 6; fungemia in 3; central nervous system (CNS) in 2; spleen in 2); in 7 (4%) the sinus focus spread to adjacent focus (CNS in 3, skin 3, eye 2, and bone 2); in 1 case a CNS focus plus adjacent bone was identified, and in 1 case the patient presented fungemia plus skin involvement.

### Diagnosis

A positive culture was obtained in 105 (59.6%) cases, corresponding to non-invasive respiratory samples in 49 (46.7%) cases (from tracheal aspirate in 45 and sputum in 4), BAL in 30 (28.6%), sinus secretion in 23 (21.9%), skin in 5 (4.8%),

**Table 1. General characteristics of children and adults with invasive mold diseases.**

| | Total n = 176 (%) | Children n = 41 (%) | Adults n = 135 (%) | p |
|---|---|---|---|---|
| Male gender | 100 (56.8) | 17 (41.5) | 83 (61.5) | 0.03 |
| Age (years old), median [ICR] | 53[18-65] | 11 [4–13] | 56.5 [46-67] | 0.00 |
| Immunosuppression* | 109 (61.9) | 39 (95.1) | 70 (51.8) | 0.00 |
| Cancer** | 78 (44.3) | 30 (73.2) | 48 (36) | 0.00 |
| Leukemia/lymphoma | 61 (34.6) | 22 (53.6) | 39 (28.9) | 0.004 |
| Neutropenia (≤100 cell/mm$^3$) | 44 (25) | 25 (61) | 19 (14) | 0.02 |
| HCT | 22 (12.5) | 14 (34.1) | 8 (5.9) | 0.00 |
| Solid organ transplantation | 8 (4.5) | 0 | 8 (5.9) | 0.20 |
| Corticosteroids used for more than 14 days | 38 (21.5) | 17 (41.4) | 21 (15.6) | 0.00 |
| Immunosuppressive drugs*** | 37 (21) | 13 (31.7) | 24 (17.8) | 0.08 |
| HIV infection | 5 (2.9) | 0 | 5 (3.7) | 0.59 |
| Viral pneumonia | 71 (43.3) | 8 (20) | 63 (46.6) | 0.00 |
| Diabetes mellitus | 50 (28.4) | 3 (7.3) | 47 (34.8) | 0.00 |
| CKD | 15 (8.5) | 0 | 15 (11.1) | 0.02 |
| COPD | 10 (5.7) | 0 | 10 (7.4) | 0.55 |

*due to any condition (cancer, HCT, drugs, etc.)

**hematologic malignancies and solid organ cancer

***others than corticosteroids

HCT: hematopoietic cell transplantation

CKD: Chronic kidney conditions (Creatinine clearance < 60 ml/min)

COPD: Chronic obstructive pulmonary disease

blood in 4 (3.8%) and other samples in 22 (20.9%), with a total of 130 mold identified (Table 2). The most frequent mold was *Aspergillus* spp. with 93 (71.5%) isolates, followed by Mucorales in 19 (14.6%), then *Fusarium* spp. 8 (6.1%), *Alternaria* spp. 3 (2.3%), *Scedosporium* spp. 2 (1.5%), and others in 5 (3.8%).

Serum GM was performed in 133 cases, with an average of 3.4 samples per patient; in 59 cases, at least one result ≥ 0.5, with an average of 2.9. Serum BDG was performed only in 32 cases, with an average of 1.6 samples per patient, resulting in >80 pg/mL in 11 cases, with an average of 356.8 pg/ml. In 21 cases, a positive result of the molecular biology technique was obtained, and in 46 cases, the histological study demonstrated the presence of hyphae in the tissue. BAL was performed in 96 cases, staining showed hyphae in 8 cases, and fungal culture was positive in 30 (29 *Aspergillus* spp. and one Mucoral). On average, 1.2 samples from BAL were processed for GM per patient; in 56 cases, it was > 1.0, with an average value of 5.25. Therefore, in 30 cases, the microbiological support for aspergillosis diagnosis was based on the positive serum-GM and/or BAL-GM.

Of the 135 cases with pulmonary involvement, chest tomography showed a mixed pattern in 100 and a single imaging pattern in 35 cases. A consolidation image was evidenced in 79, nodules in 59 cases (27 with halo-sign), ground glass in 47, cavitations in 26, pleural effusions in 25, tree in the bud in 14, bronchiectasis in 7, and air-crescent nodules in 5. In the 38 cases with sinus involvement, the tomography showed sinusitis in 27, bone involvement in 8, and CNS invasion in 4. In 12 cases, CNS involvement was suspected by images, with focal lesions in 10 cases and meningeal contrast uptake in two cases.

According to the certainty of diagnosis [32,33], 53 (30.1%) cases were proven, 98 (55.7%) probable and 25 (14.2%) possible.

**Table 2. One hundred and thirty filamentous fungi identified in 105 invasive mold diseases cases.**

| Fungal order | Fungal genus/species | n | % |
|---|---|---|---|
| Eurotiales n = 94 (72.3%) | *Aspergillus fumigatus* | 46 | 35.4 |
| | *Aspergillus niger* | 16 | 12.3 |
| | *Aspergillus flavus* | 11 | 8.5 |
| | *Aspergillus* spp. | 8 | 6.2 |
| | *Aspergillus terreus* | 7 | 5.4 |
| | *Aspergillus nidulans* | 3 | 2.3 |
| | *Aspergillus lentulus* | 2 | 1.5 |
| | *Paecilomyces* spp. | 1 | 0.8 |
| Mucorales n = 19 (14.6%) | *Mucor* spp. | 9 | 6.9 |
| | *Rhizopus* spp. | 9 | 6.9 |
| | *Lichtheimia* spp. | 1 | 0.8 |
| Hypocreales n = 9 (6.9%) | *Fusarium solani complex* | 6 | 4.6 |
| | *Fusarium* spp. | 2 | 1.5 |
| | *Sarocladium kiliense* | 1 | 0.8 |
| Pleosporales n = 5 (3.8%) | *Alternaria* spp. | 3 | 2.3 |
| | *Curvularia* spp. | 1 | 0.8 |
| | *Ulocladium* spp. | 1 | 0.8 |
| Microascales n = 3 (2.3%) | *Scedosporium* spp. | 2 | 1.5 |
| | *Lomentospora prolificans* | 1 | 0.8 |

Among the 151 IMDs cases with microbiological criteria, aspergillosis was the most frequently IMD diagnosed for both children and adults, being diagnosed in 114 (75.5%) cases (23 in children and 91 cases in adults), followed by mucormycosis in 16 (10.6%) cases (1 in children and 15 in adults), then fusariosis in 7 (5,3%) cases (3 in children and 4 in adults). In summary, fusariosis in children and mucormycosis in adults were the second most frequent IMDs after aspergillosis. Aspergillosis coinfection with other mold was diagnosed in 5 (3.3%) cases, 3 with Mucorales (1 child and two adults), 1 with *Alternaria* spp., and 1 with *Fusarium* spp. (both cases in children).

The most frequent manifestation was fever in 144 (81.8%) cases, respiratory symptoms in 107 (60.8%) cases, sinus manifestations in 31 (17.6%), neurologic signs in 17 (9.7%) and skin findings in 15 (8.5%). Table 3 shows the clinical manifestation in 114 cases of aspergillosis, 16 mucormycosis, and 7 fusariosis cases (fungal coinfection cases are not included). Respiratory symptoms were more frequent in aspergillosis, rhinosinusal/neurological signs in mucormycosis, and rhinosinusal/cutaneous in fusariosis. No statistically significant differences were observed comparing symptoms and signs among aspergillosis, mucormycosis, and fusariosis cases in children vs. adult patients.

## IMD cases associated with viral pneumonia

Seventy-one (40.3%) cases were associated to viral pneumonia, 60 (34.1%) to SARS-CoV-2 infection, 5 (2.8%) to respiratory syncytial virus (RSV) (1 RSV plus influenza and 1 RSV plus parainfluenza), 4 (2.3%) to influenza (3 influenza B and one influenza A) and 2 (1.1%) to rhinovirus. Regarding the type of IMD, in 62 (87.3%) cases were aspergillosis, in 2 (2.8%) mucormycosis, in 2 (2.8%) *Scedosporium* spp. infection, and in 2 (2.8%) fungal coinfections (*Aspergillus* spp. plus Mucorales and *Aspergillus* spp. plus *Alternaria* spp.), and in 3 (4.2%), the fungus was not identified. Regarding the certainty of diagnosis, 33 of the 55 cases of CAPA, 42 (76.2%) were probable, and 13 (23.6%) were possible. In contrast, the COVID-19-associated mucormycosis and the aspergillosis/mucormycosis coinfection corresponded to proven cases due to the biopsy showing invasive hyphae in the tissue.

**Table 3. Clinical manifestation in aspergillosis, mucormycosis and fusariosis cases.**

| | Aspergillosis n = 114 (%) | Mucormycosis n = 16 (%) | Fusariosis n = 7 (%) |
|---|---|---|---|
| **Fever** | **92 (80.7)** | **13 (81.2)** | **6 (85.7)** |
| **Respiratory** | **82 (89.1)** | **4 (25)** | **1 (14.3)** |
| Respiratory failure | 32 (28.1) | 1 (6.2) | |
| Cough | 26 (22.8) | 2 (12.5) | 1 (14.3) |
| Dyspnea | 16 (14) | 1 (6.2) | |
| Hemoptysis | 1 (1.2) | | |
| Other | 7 (6.1) | | |
| **Rhinosinusal** | **11 (9.6)** | **10 (62.5)** | **3 (42.9)** |
| Necrotic lesion (palate and/or nasal) | 3 (2.6) | 5 (31.2) | 3 (42.9) |
| Facial pain | 3 (2.6) | 2 (12.5) | |
| Periorbital cellulitis | 3 (2.6) | 1 (6.2) | |
| **Neurologic** | **6 (5.3)** | **6 (37.5)** | **1 (14.3)** |
| Headache | 2 (1.7) | 3 (18.7) | 1 (14.3) |
| Altered level of consciousness | 2 (1.7) | 2 (12.5) | |
| Focal neurologic deficit | 2 (1.7) | 1 (6.2) | |
| **Cutaneous** | **4 (3,5)** | **5 (31.1)** | **3 (42.9)** |
| Nodules | 2 (1.7) | | 2 (28.6) |
| Papules | 2 (1.7) | 2 (12.5) | |
| Cellulitis | | 2 (12.5) | |
| Necrotic lesion | | 1 (6.2) | 1 (14.3) |

## Breakthrough IMD

Fifty-nine (29%) IMD cases were diagnosed in patients who had received at least one dose of antifungal during the 14 days before the diagnosis of IMD. However, only 30 (17%) cases received antifungals for seven or more days and were considered a b-IMD. Fluconazol was the antifungal more frequently prescribed in 18 (60%) cases, followed by voriconazole in 6 (20%), liposomal amphotericin in 4 (13.3%), echinocandins in 3 (10%), posaconazole in 1 (3.3%). The antifungals were used for 38 ± 85 days. Prophylaxis was the more frequent prescription reason in 18 (60%) cases (primary prophylaxis in 15 and secondary in 3), targeted therapy in 6 (20%), empirical treatment in 4 (13.3%), and in 2 (6.7%) cases the reason of prescription was not registered. The b-IMD were aspergillosis in 14 (46.7%) cases, mucormycosis in 6 (20%), fusariosis in 1 (3.3%), *Alternaria* spp., *Lomentospora prolificans* and *Curvularia lunata* in 1 case each one, also 1 case of *A. flavus* and *F. solani* coinfection, and in 5 (16.7%) the type of b-IMD was not identified.

## Antifungal susceptibility

Not all strains were viable when we proceeded with the susceptibility test. Table 4 shows the susceptibility of 43.5% of *A. fumigatus* (20 of 46 strains), 37.5% of *A. niger* (6 of 16), and 81.8% of *A. flavus* (9 of 11). No triazole resistance was observed. In fact, all strains presented MICs under the ECV breakpoint for itraconazole, posaconazole and voriconazole. On the contrary, 5% of *A. fumigatus* and 11% of *A. flavus* were non-wild types of amphotericin B. No isolate of *A. niger* presented amphotericin MICs above the ECV. Instead, a high percentage presented MICs above the ECVs for caspofungin.

## Antifungal therapy

One hundred and seventy-one (97.2%) IMDs cases received antifungals with anti-mold activity, and 5 cases did not (2 of them due to postmortem diagnosis, 2 were too severely sick to begin new therapies, and one was not registered the reason). The

**Table 4.** Minimal inhibitor concentration (MIC) and epidemiological cut-off value (ECV) for amphotericin, caspofungin, and azoles in 35 *Aspergillus* spp.

| | Antifungal | MIC range ug/mL | MIC 50 | MIC 90 | ECV | % over ECV |
|---|---|---|---|---|---|---|
| *Aspergillus fumigatus* n=20 | Amphotericin B | 0.5 - 4 | 2 | 2 | 2 | 5 |
| | Caspofungin | 0.06 ->8 | >8 | >8 | 0.5 | 95 |
| | Itraconazole | 0.015–0.5 | 0.12 | 0.5 | 1 | 0 |
| | Posaconazole | 0.015–0.25 | 0.3 | 0.12 | NA | NA |
| | Voriconazole | 0.12-1 | 0.25 | 0.5 | NA | NA |
| *Aspergillus flavus* n=9 | Amphotericin B | 1 a 8 | 2 | 4 | 4 | 11 |
| | Caspofungin | 8->8 | 8 | >8 | 0.5 | 100 |
| | Itraconazole | 0.03 a 5 | 0.06 | 0.25 | 1 | 0 |
| | Posaconazole | 0.03 a 5 | 0.06 | 0.25 | 0.5 | 11 |
| | Voriconazole | 0.25 a 1 | 0.5 | 0.5 | 2 | 0 |
| *Aspergillus niger* n=6 | Amphotericin B | 1 a 2 | 2 | 2 | 2 | 0 |
| | Caspofungin | 0.06 a>8 | 8 | >8 | 0.25 | 83 |
| | Itraconazole | 0.06 a 1 | 0.25 | 0.5 | 4 | 0 |
| | Posaconazole | 0.06-0.25 | 0.12 | 0.12 | 2 | 0 |
| | Voriconazole | 0.5 a 1 | 0.5 | 0.5 | 2 | 0 |

NA: Not applicable

initial antifungal prescription was voriconazole in 105 (61.4%) cases, liposomal amphotericin B in 35 (20.5%), combination antifungals in 19 (11.1%) (voriconazole plus lipid formulation of amphotericin in 16 and voriconazole plus echinocandin in 3), isavuconazole in 5 (2.9%), amphotericin deoxycholate in 4 (2.3%), echinocandins in 2 (1.2%). For aspergillosis, voriconazole was the most frequent prescription in 79 (69.3%) cases, followed by liposomal amphotericin [in combination with voriconazole 10 (8.8%) or as monotherapy 8 (7%)] and in 2 (1,8%) cases voriconazole plus an echinocandin; in mucormycosis liposomal amphotericin was the most frequent prescription 13 (81.5%) or with a triazole in 2 (12.5%) cases; and in fusariosis the more used therapy was amphotericin with voriconazole combination in 4 (57.1%) cases, followed by amphotericin or voriconazole in 2 (28,6%) and 1 (14.3%) case respectively. Concerning the duration of antifungal therapy, it was 60.6±77.7 days; in aspergillosis, cases was 53.3±79.6 days; in mucormycosis, 69.5±66.9 days, and in fusariosis 107±89.3 days.

Of the patients who received voriconazole, at least one plasma level was measured in 94 (77.7%) cases. Surgical debridement was performed in 41 (23.3%) cases: 17 (14.9%) cases of aspergillosis, 13 (81.3%) cases of mucormycosis, 5 (71.4%) cases of fusariosis, 2 (100%) cases of *Alternaria* spp. infection, and 2 (40%) cases of coinfections.

## Outcomes

Table 5 shows the LOS, the complete response to therapy, and the survival rate in children and adults for total IMDs, aspergillosis, mucormycosis, and fusariosis. Overall survival from the diagnosis of IMD was 58.5% at 30 days (103 cases), 45.5% at 90 days (80 cases), and 32.4% at 180 days (57 cases). Comparing IMD in children and adult cases, a higher survival rate is observed in children vs. adults at 30, 90, and 180 days. No statistically significant differences were observed in comparing children vs. adults regarding outcomes among aspergillosis, mucormycosis, and fusariosis cases.

## Discussion

This is the most extensive study carried out in our country to determine the epidemiology of IMDs in children and adults. It evidences an overall incidence of 0.4 per 1,000 admissions, with aspergillosis being the most frequent infection, with an incidence of 0.2, followed by mucormycosis, 0.03, and fusariosis, 0.002 per 1,000 admissions.

Table 5. Clinical outcomes: Length of stay (LOS), complete response (CR) at 6 and 12 weeks, and survival rate (SR) at 30-90-180 days for the total of 176 invasive mold diseases cases and for the aspergillosis, mucormycosis, and fusariosis cases, comparing children and adults.

| | Total IMD | | | | Aspergillosis | | | | Mucormycosis | | | | Fusariosis | | | |
|---|---|---|---|---|---|---|---|---|---|---|---|---|---|---|---|---|
| | Total n=176 | Children n=41 | Adults n=135 | p | Total n=114 | Children n=23 | Adults n=91 | p | Total n=16 | Children n=1 | Adults n=15 | p | Total n=7 | Children N=3 | Adults n=4 | p |
| LOS, days (median [IQR]) | 52 [25-109] | 67.5 [33-151] | 51 [23-106] | 0.08 | 44.5 [23-81] | 61 [35-81] | 36 [21-81] | 0.05 | 53 [32-154] | 45 | 62 [32-154] | 0.82 | 51.5 [50–60] | 139.5 [33-246] | 51.5 [50.5-56] | 1.00 |
| CR at week 6 n (%) | 40 (22.7) | 6 (14.6) | 34 (25.2) | 0.16 | 23 (20.2) | 3 (13) | 20 (22) | 0.40 | 4 (25) | 0 | 4 (26.7) | 1.00 | 0 | 0 | 0 | |
| CR at week 12 n (%) | 58 (33) | 15 (37) | 43 (32) | 0.58 | 34 (29.8) | 9 (39) | 25 (27,5) | 0.31 | 5 (31.2) | 0 | 5 (33) | 1.00 | 0 | 0 | 0 | |
| 30 days SR n (%) | 103 (58.5) | 34 (82.9) | 69 (51.1) | 0.00 | 55 (48.3) | 14 (60.9) | 41 (45) | 0.24 | 9 (56.3) | 1 (100) | 8 (53.3) | 1.00 | 5 (71.4) | 2 (66.7) | 3 (75) | 1.00 |
| 90 days SR n (%) | 80 (45.5) | 25 (61) | 55 (40.7) | 0.03 | 47 (41.2) | 14 (60.9) | 33 (36.2) | 0.06 | 8 (50) | 0 | 8 (53.3) | 1.00 | 4 (57.1) | 2 (66.7) | 2 (50) | 1.00 |
| 180 days SR n (%) | 57 (32.4) | 17 (41.5) | 40 (29.6) | 0.18 | 27 (23.7) | 9 (39.1) | 18 (19.8) | 0.06 | 7 (43.7) | 0 | 7 (46.7) | 1.00 | 3 (42.9) | 2 (66.7) | 1 (25) | 0.49 |

IQR: Interquartile range

Interestingly, the incidences of aspergillosis and mucormycosis are close to those indicated in the Alvarez et al. study to estimate the burden of fungal infection in Chile, in which an 8.5-fold superiority was calculated in the rate of aspergillosis over mucormycosis [23]. Similar to our results, European, USA and Southeast Asia reports highlight that aspergillosis is the most frequent IMD; mucormycosis usually ranks second in incidence, followed by fusariosis and other filamentous [37–39]. In particular, in different Latin American countries, aspergillosis has been considered the most frequent IMD in Argentina [40], and in hematological patients in Brazil [41]. However, in Brazil, fusariosis is the second most frequent in HCT and hematologic malignancies [42]. A specific finding in the children vs. adults comparison was that the second most frequent infection was fusariosis vs. mucormycosis, respectively. This observation had not been previously reported in epidemiological studies of children [43]. We interpreted that the frequency of fusariosis in children may be linked to the higher burden of immunosuppression due to hematological diseases and neutropenia present in the patients included. In fact, hematologic malignancies and solid tumors requiring intensive immunosuppression are the main conditions related to fusariosis in children [44]. On the other hand, the higher frequency of mucormycosis in adults could be associated with the higher incidence of poorly controlled diabetes in this group. Diabetes and diabetic cetoacidosis are known risk factors for mucormycosis, in a global review about mucormycosis, is remarked that the high incidence reported from India is probably explained for their high rate of diabetes, on the contrary in USA or Europe, hematological malignancies are the conditions more frequently associated to mucormycosis [45].

These epidemiological observations should be considered in the clinical approach and therapeutic decision-making process, recognizing that aspergillosis is the most frequent IMD etiology, but also recognizing that mucormycosis and fusariosis are the next most prevalent identified molds.

Among the underlying conditions, those related to the immune system compromise were the more frequent, such as leukemia/lymphoma, neutropenia, and the use of corticosteroids, coinciding with the most reported conditions [46]. However, HCT and SOT were not found among the most common baseline conditions, which may reflect the heterogeneity of the type of patients in the participating centers and the antifungal prophylaxis practices. On the other hand, the high frequency of viral pneumonia should be highlighted, which reflects the burden of COVID-19 during the study period, as well as diabetes, CKD, and COPD among chronic conditions in the adult population. Concerning the relatively low percentage of COPD, it could account for low suspicion of aspergillosis in this condition despite its association having been reported more frequently during the last decade [47].

The more frequent clinical manifestation was fever. However, it also showed the relevance of having a high level of clinical suspicion to insist on looking for IMDs, considering that up to 18.2% of the patients did not present fever. This fact is also described in a report of 116 invasive aspergillosis among non-neutropenic patients, in which fever was present in 57.8% of cases [48]. As a consequence, IMDs should always be considered in at-risk patients, and the diagnostic approach should be triggered using different clinical and laboratory elements.

Regarding the role of microbiology, biomarkers, imaging, BAL, and molecular techniques in diagnosis, the results confirm the need for different diagnostic tests to achieve the highest accuracy. The combination of GM and polymerase chain reaction-based *Aspergillus* DNA detection has been helpful for the early diagnosis of invasive infection in high-risk hematological patients [49]. However, the costs of this strategy must be considered, especially if they are part of a pre-emptive approach in which samples must be repeated 2 or 3 times a week.

In our series, 17% of the cases were b-IMD; most were receiving fluconazole, which has no activity against molds, but 46.6% received antifungal with anti-mold activity. As reported in the breakthrough infections series, aspergillosis and mucormycosis are the more frequent mold etiologies [50]. Also, we observed IMD cases without mycological confirmation, representing the challenge of obtaining specific diagnostics in these patients, considering the lower performance of biomarkers and cultures. It remarks the importance of conducting a completed workup in cases when b-IMD is suspected, including biopsies, if clinically indicated [51].

Almost all patients received antifungal therapy; the most frequent prescriptions followed those recommended in clinical guidelines, with voriconazole predominating in aspergillosis [52], and liposomal amphotericin B in mucormycosis [53]. In change, for fusariosis, a combination of voriconazole plus liposomal amphotericin B was the most frequent therapy, which differs from the guidelines recommending voriconazole as first-line therapy [54]. The lower use of isavuconazole in aspergillosis could be explained by the fact that it was incorporated more recently in our country. The measurement of voriconazole plasma levels is a widespread practice in the participating centers and should be maintained considering its narrow therapeutic margin and the difficulties observed in children in dosing adequately [55].

According to Macedo et al., until October 2020, *Aspergillus* spp. resistance to triazoles had been reported from Argentina, Brazil, Colombia, and Peru but not from Bolivia, Chile, Ecuador, Guyana, Paraguay, and Uruguay. This indicates that azole resistance is emerging in Latin America and should be monitored [26]. Our results show no triazole resistance was demonstrated during this period among the *Aspergillus* spp. isolated from participating centers in Chile.

The triazole resistance in *Aspergillus spp.* is related to the poorly controlled use of antifungals, both clinically and environmentally, such as pesticides. The mean MIC values for voriconazole and posaconazole from environmental strains are lower than those from clinical isolates [56]. The molecular mechanisms of resistance to *A. fumigatus* mainly involve substitution in the azole target site CYP51A and/or overexpression of this gene. However, other resistance mechanisms have also been described, and intrinsically resistant cryptic *Aspergillus* species have been reported in the clinic [26]. Regarding the triazole resistance of other *Aspergillus* species, less information has been published and could be much higher than what is currently known [57]. A large screening study of 1789 *Aspergillus* strains showed that the frequency of non-wild type *A. flavus* isolates for itraconazole, voriconazole, and posaconazole was 0.8%, 1.7%, and 5.1%, respectively [58]. So, performing a routine in vitro susceptibility testing of all *Aspergillus* isolates for clinical and epidemiological purposes would be advisable, at least in reference centers.

Related to polyenes, we found that 5% and 11% of the strains of *A. fumigatus* and *A. flavus*, respectively, were the non-wild types to amphotericin B, but in any case, this observation does not mean resistance. Polyenes resistance is sporadic in other species of *Aspergillus* different than *A. terreus* [56]. However, some *A. flavus*, with mutations in the *cyp51C* gen, can present intrinsic resistance to this antifungal agent [57]. Additionally, it should be noted that non-wild type strains of *A. niger* were found to amphotericin B, according to the CLSI M57S standard. However, according to the EUCAST clinical breakpoints, 83% would not be sensitive to amphotericin B. However, the strains did not show values above the ECVs to voriconazole, to which, clinically, *A. niger* responds adequately [58].

A high percentage of the *Aspergillus* spp. isolates in our study presented MICs above the ECVs for caspofungin, with MIC 50 and MIC 90 being 8 ug/mL. Mutations in *A. fumigatus* occur in the glucan synthase gene AfFKS1, which, together with increased chitin production, develops *Aspergillus* resistance to echinocandins [59]. Based on our data, caspofungin should not be considered an antifungal choice in aspergillosis.

The survival rate observed in our series is comparable to other published [37]. Access to appropriate antifungal therapies, their adequate dosage, and associated surgical debridement in cases of mucormycosis influence these results. The better survival rate observed in children vs. adults could be related to the higher frequency of severe neutropenia, as a transitory condition, compared to the higher frequency of chronic conditions among adults. However, other factors could influence this outcome.

We recognize some limitations in our study: the centers included were heterogeneous in the populations of patients at risk, prevention policies, availability of protected environment units, and the use of antifungal prophylaxis, as well as the suspicion and diagnosis protocols. Additionally, including 14% of possible cases could lead to overestimating the prevalence of IMDs in our country. However, considering that the study seeks to know the epidemiological reality, it is valid to incorporate centers with differences in infrastructure, diagnostic capacities, and management. The challenge will be to advance toward a national clinical protocol and contribute to developing a regional guideline to standardize the best practices. Including a pandemic period undoubtedly influences the results, considering that 60 cases were associated with COVID-19. However, a non-pandemic year was also included, with 11 cases related to viral pneumonia from another agent, highlighting the importance of the relationship between respiratory viruses and IMDs. Another limitation was that not all *Aspergillus* isolated were possible to include for the antifungal susceptibility test. Other limitation of our analysis was the unavailability of data regarding the number of patients with underlying conditions at each participating center during the study period. Consequently, it was not possible to calculate the incidence rate of IMD for each specific condition (e.g., leukemia, diabetes, transplantation, etc.). Future epidemiological studies should include this information to enable a more accurate and condition-specific assessment of IMD incidence.

In summary, 176 cases of IMDs were reported from 11 reference centers in Chile; Aspergillosis was the most frequently identified IMD in the total group and in each age group. The second cause was fusariosis in children and mucormycosis in adults. Nearly 40% of the IMDs were associated with respiratory viruses, accounting for the impact of COVID-19 on the emergence of superinfection by *Aspergillus* spp. Despite many patients starting antifungal therapy, the survival rate was poor. It is advisable to start an active surveillance of IMDs in Chile and verify the emergence of resistance of *Aspergillus* spp. over time.

## Supporting information

**S1 File. Molds database Chile Aug 2025.**
(XLSX)

## Acknowledgments

Other members of the Chilean Invasive Mycosis Network, investigators and laboratory technicians, involved in this protocol: Elizabeth Barthel, Leonor Jofré, Loreto Twele, Pilar Rodríguez, María Luz Endeiza, Marcela Rabello, Raúl Quintanilla, Isabel Briceño, Leonardo Siri, Rodrigo Ahumada, Sebastián Barría, Carla Concha, Ignacio Delama, Claudia Cortes, Mabel Aylwin, Jeanette Dabanch, Fernanda Cofré, Laura Bahamones, Francisco Silva, Alfonso Guzmán, Ricardo Morales, Gloria Marín, Leonardo Chanqueo, Francisca Valdivieso, Loriana Castillo, Ernesto Paya, Alejandro Joyas, María Luis Rioseco, Vijna Illesca, Patricio Godoy, Mónica Lafourcade, Lorena Porte, Pamela Rojas, Bélgica Barraza, Freddy Roach, Gerardo Peralta, Francisca Plummer

## Author contributions

**Conceptualization:** Ricardo Rabagliati, Dona Benadof, Luis Thompson, Patricia Garcia, Cecilia Tapia, Maria Elena Santolaya.

**Data curation:** Dona Benadof, Patricia Garcia, Tamara Gonzalez, Cecilia Vizcaya, Andres Soto, Catalina Gutierrez, Lorena Rodriguez, Yenis Labraña, Cecilia Tapia, Marcela Zubieta, Martin Lasso, Valentina Gutierrez, Veronica Contardo, Yocelyn Castillo, Mario Calvo, Karen Ducasse, Romina Valenzuela, Maria Elena Santolaya.

**Formal analysis:** Ricardo Rabagliati, Luis Thompson, Cecilia Vizcaya, Andres Soto, Catalina Gutierrez, Yenis Labraña, Cecilia Tapia, Marcela Zubieta, Martin Lasso, Valentina Gutierrez, Veronica Contardo, Yocelyn Castillo, Romina Valenzuela, Maria Elena Santolaya.

**Funding acquisition:** Dona Benadof.

**Investigation:** Ricardo Rabagliati, Dona Benadof, Patricia Garcia, Tamara Gonzalez, Cecilia Vizcaya, Andres Soto, Catalina Gutierrez, Lorena Rodriguez, Yenis Labraña, Cecilia Tapia, Marcela Zubieta, Martin Lasso, Valentina Gutierrez, Veronica Contardo, Yocelyn Castillo, Mario Calvo, Karen Ducasse, Romina Valenzuela, Maria Elena Santolaya.

**Methodology:** Ricardo Rabagliati, Dona Benadof, Luis Thompson, Patricia Garcia, Tamara Gonzalez, Cecilia Tapia, Romina Valenzuela, Maria Elena Santolaya.

**Project administration:** Ricardo Rabagliati, Romina Valenzuela.

**Software:** Romina Valenzuela.

**Supervision:** Ricardo Rabagliati, Patricia Garcia, Maria Elena Santolaya.

**Validation:** Ricardo Rabagliati, Dona Benadof, Patricia Garcia, Tamara Gonzalez, Lorena Rodriguez, Mario Calvo, Romina Valenzuela, Maria Elena Santolaya.

**Visualization:** Luis Thompson, Lorena Rodriguez, Romina Valenzuela.

**Writing – original draft:** Ricardo Rabagliati, Dona Benadof, Luis Thompson, Patricia Garcia, Cecilia Vizcaya, Andres Soto, Catalina Gutierrez, Cecilia Tapia, Marcela Zubieta, Martin Lasso, Valentina Gutierrez, Veronica Contardo, Mario Calvo, Karen Ducasse, Romina Valenzuela, Maria Elena Santolaya.

**Writing – review & editing:** Ricardo Rabagliati, Luis Thompson, Maria Elena Santolaya.

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
