## [Decision Letter · Decision Letter 0]

8 Aug 2024

Dear Dr. Rabagliati,

Thank you for submitting your manuscript to PLOS ONE. After careful consideration, we feel that it has merit but does not fully meet PLOS ONE’s publication criteria as it currently stands. Therefore, we invite you to submit a revised version of the manuscript that addresses the points raised during the review process.

We look forward to receiving your revised manuscript.

Kind regards,

Nosheen Nasir

Academic Editor

PLOS ONE

Additional Editor Comments (if provided):

Reviewers' comments:

Reviewer's Responses to Questions

**Comments to the Author**

1. Is the manuscript technically sound, and do the data support the conclusions?

Reviewer #1: Yes

Reviewer #2: Partly

2. Has the statistical analysis been performed appropriately and rigorously?

Reviewer #1: Yes

Reviewer #2: N/A

3. Have the authors made all data underlying the findings in their manuscript fully available?

Reviewer #1: Yes

Reviewer #2: Yes

4. Is the manuscript presented in an intelligible fashion and written in standard English?

Reviewer #1: Yes

Reviewer #2: Yes

Reviewer #1: Invasive mold disease (IMD) is a severe complication which affects immunocompromised and immunocompetent patients. In this manuscript, the authors have investigated the epidemiological and clinical features of IMD in patients specifically from Chile between the time period of 2019 to 2021. Some suggestions regarding improving or modifying the manuscript is as follows :

1. Page 4: Please re write the abstract since it seems rather disconnected and abrupt.

2. Page 4: In the abstract, the words "The objective" is in italics.

3. Page 4: In the results --What is immunosuppression in the pediatric group, the result of?

4. Page 8: Could you explain in detail what the patient conditions were, for data analysis?

5. The numerical data is not consistent throughout the paper --in some places the numbers are given as digits and in other places, the numbers are spelt out.

6. Please provide the full forms of all the abbreviations mentioned.

7. Have you looked into Crytococcosis, considering that WHO has mentioned it as an emerging disease?

Reviewer #2: Report on Review of "A prospective, multicenter study of invasive fungal disease caused by molds in children and adults in Chile"

The submitted manuscript offers valuable insights into the epidemiology of IMDs in Chile. However, there are several areas where the manuscript could be improved to enhance clarity and comprehensiveness.

Terminology: The authors should consistently use the plural form "Invasive mold diseases (IMDs)" throughout the manuscript, as IMDs encompass a broad spectrum of diseases caused by various mold species.

Study Design:

-Please clarify the composition of the control group in this prospective observational study. How were the control subjects selected and compared to the case group?

-A detailed description of the inclusion and exclusion criteria for both groups would strengthen the study design.

Methodology:

-A more comprehensive description of the sample collection, processing, and culture techniques is needed.

-The rationale for limiting antifungal susceptibility testing to Aspergillus Spp. should be explained.

Results:

-The clinical manifestations of IMDs should be presented separately for each type of infection to provide a clearer understanding of the disease spectrum.

-The specific molecular technique used for the identification of mold species should be stated.

-The results for children and adults should be presented separately, especially for the proven, probable, and possible categories.

-A more detailed analysis of the antifungal treatment regimens would be beneficial. For each disease, please specify which antifungal agents were used and for what duration.

Discussion:

Although the manuscript centers on IMDs, it disproportionately emphasizes Aspergillus. The data, however, indicates that fusariosis and mucormycosis are prevalent infections among pediatric and adult populations. The discussion should give more weight to these findings and their implications for clinical management.

**Do you want your identity to be public for this peer review?** For information about this choice, including consent withdrawal, please see our Privacy Policy

Reviewer #1: No

Reviewer #2: No

---

## [Author Response · Author response to Decision Letter 1]

21 Feb 2025

RESPONSE TO REVIEWERS

Dr. Nosheen Nasir

Academic Editor

PLOS ONE

Dear Dr Nasir.

Thank you for your letter referring to the revision of our manuscript PONE-D-24-19295. We also thank the reviewers for their diligent revision and comments that will certainly improve the manuscript. We will address the reviewer comments one by one, pointing out when changes to the original manuscript were made following the reviewers’ suggestions.

Reviewers' comments:

Reviewer #1: Invasive mold disease (IMD) is a severe complication which affects immunocompromised and immunocompetent patients. In this manuscript, the authors have investigated the epidemiological and clinical features of IMD in patients specifically from Chile between the time period of 2019 to 2021. Some suggestions regarding improving or modifying the manuscript is as follows:

1. Page 4: Please re write the abstract since it seems rather disconnected and abrupt.

Response. We re wrote the manuscript in a more comprehensive manner.

2. Page 4: In the abstract, the words "The objective" is in italics.

Response. We delete the italics

3. Page 4: In the results --What is immunosuppression in the pediatric group, the result of?

Response. Thanks for this suggestion. We change the term immunosuppression by the following phrase in the abstract: Cancer, neutropenia and hematopoietic cell transplantation were the most frequently associated conditions in the pediatric group, and we added the same information in the Results section (page 12)

4. Page 8: Could you explain in detail what the patient conditions were, for data analysis?

Response. The immunosuppression-related conditions were cancer, neutropenia and HCT, as was explained in the abstract and results.

5. The numerical data is not consistent throughout the paper --in some places the numbers are given as digits and in other places, the numbers are spelt out.

Response. We change all numbers to digits, with the exception of numbers that are at the beginning of a paragraph or immediately after a period.

6. Please provide the full forms of all the abbreviations mentioned.

Response. The full forms were provided previous to abbreviations throughout the manuscript

7. Have you looked into Crytococcosis, considering that WHO has mentioned it as an emerging disease?

Response. Thanks for this suggestion. We don’t inform about Cryptococcus because this paper aimed to describe infections caused by molds, not yeasts.

Reviewer #2: Report on Review of "A prospective, multicenter study of invasive fungal disease caused by molds in children and adults in Chile" The submitted manuscript offers valuable insights into the epidemiology of IMDs in Chile. However, there are several areas where the manuscript could be improved to enhance clarity and comprehensiveness.

1.Terminology: The authors should consistently use the plural form "Invasive mold diseases (IMDs)" throughout the manuscript, as IMDs encompass a broad spectrum of diseases caused by various mold species.

Response. We changed IMD to IMDs throughout the manuscript

2.Study Design: Please clarify the composition of the control group in this prospective observational study. How were the control subjects selected and compared to the case group?

Response. This study only included IMDs cases. Controls were not included

3. A detailed description of the inclusion and exclusion criteria for both groups would strengthen the study design.

Response. We incorporated a IMD case definition in the Patients and methods section, to better describe the cases included (Page 10)

4. Methodology: A more comprehensive the sample description of collection, processing, and culture techniques is needed.-The rationale for limiting antifungal susceptibility testing to Aspergillus Spp. should be explained.

Response. Thanks for these important suggestions. We improve the Laboratory procedures paragraph including collection, culture, identification (Page 9-10), and the explanation that not all the Aspergillus strains were viable for the susceptibility test (Page 17), also we include a commentary that it is a limitation of our study (Page 24)

5. Results: The clinical manifestations of IMDs should be presented separately for each type of infection to provide a clearer understanding of the disease spectrum.

Response.

Thanks again for the commentary. We include a new table (Table 3) with more information about the clinical manifestation in aspergillosis, mucormycosis and fusariosis cases, describing in the manuscript the more relevant clinical differences among them (Page 15)

6. The specific molecular technique used for the identification of mold species should be stated.

Response. We include these specifications

7. The results for children and adults should be presented separately, especially for the proven, probable, and possible categories.

Response. The outcomes for children and adults, in general and for aspergillosis, mucormycosis and fusariosis are detailed in Table 5

8. A more detailed analysis of the antifungal treatment regimens would be beneficial. For each disease, please specify which antifungal agents were used and for what duration.

Response. We include these data, in special we include the antifungal therapy for aspergillosis, mucormycosis and fusariosis. (Page 18)

9. Discussion: Although the manuscript centers on IMDs, it disproportionately emphasizes Aspergillus. The data, however, indicates that fusariosis and mucormycosis are prevalent infections among pediatric and adult populations. The discussion should give more weight to these findings and their implications for clinical management.

Response. Thanks for this suggestion. We consider that inclusion of clinical manifestations and therapy in the results section (Commentaries 5 & 8), we answer this comment. In addition, we include in the discussion to consider the differential diagnosis of IMD in the clinical approach and therapeutic decision-making process (Page 19-20).

We truly hope that our responses to the reviewers and the modifications made to our manuscript make our paper suitable for publication in your Journal.

Sincerely yours,

Ricardo Rabagliati

---

## [Decision Letter · Decision Letter 1]

27 May 2025

Dear Dr. Rabagliati,

Thank you for submitting your manuscript to PLOS ONE. After careful consideration, we feel that it has merit but does not fully meet PLOS ONE’s publication criteria as it currently stands. Therefore, we invite you to submit a revised version of the manuscript that addresses the points raised during the review process.

**ACADEMIC EDITOR: Please address reviewer comments.**

We look forward to receiving your revised manuscript.

Kind regards,

Nosheen Nasir

Academic Editor

PLOS ONE

Reviewers' comments:

Reviewer's Responses to Questions

**Comments to the Author**

Reviewer #1: All comments have been addressed

Reviewer #3: (No Response)

Reviewer #4: All comments have been addressed

2. Is the manuscript technically sound, and do the data support the conclusions?

Reviewer #1: Yes

Reviewer #3: Partly

Reviewer #4: Yes

3. Has the statistical analysis been performed appropriately and rigorously?

Reviewer #1: Yes

Reviewer #3: No

Reviewer #4: Yes

4. Have the authors made all data underlying the findings in their manuscript fully available?

Reviewer #1: Yes

Reviewer #3: Yes

Reviewer #4: Yes

5. Is the manuscript presented in an intelligible fashion and written in standard English?

Reviewer #1: Yes

Reviewer #3: Yes

Reviewer #4: Yes

Reviewer #1: (No Response)

Reviewer #3: Rabagliati and colleagues collected data from over 100 pediatric and adult patients across 11 hospitals in Chile to expand the understanding of invasive mold disease (IMD) in the region. This study helps address the current gap in data regarding the severity and characteristics of IMD in Chile.

That said, I have a few comments:

Major comments:

1. Missing methods:

1.1 The conditions under which “fungal culture” was performed should be clearly described.

1.2 The source (commercial or in-house) for the “lactophenol blue stain” should be specified.

1.3 More detailed information should be provided regarding the “tube protein extraction method.”

2. The authors list the percentages of various diseases or infections among the patient cohort. It would add value to conduct hypergeometric tests comparing these frequencies to those of all hospital admissions, to determine whether these conditions are significantly enriched or associated with IMD cases.

3. In discussion:

3.1 In the sentence, “We interpreted that the frequency of fusariosis… poorly controlled diabetes in this group,” a mechanistic explanation would strengthen the argument. Providing a biological or pathophysiological basis for the association would be more informative.

3.2 The statement, “The high frequency of viral pneumonia should be highlighted… in the adult population,” would benefit from a statistical test (e.g., hypergeometric test) to assess whether this frequency is significantly associated with IMD. Since the focus of the study is on IMD, such analysis would better support the relevance of this observation.

Reviewer #4: This curret study provides readers with a vivid analysis and description of real-world data on invasive fungal infections in both adults and children in Chile. The study holds significant academic value and merits publication in PLOS ONE. However, certain aspects of the manuscript warrant further discussion and refinement.

1. What are the clinical implications of statistically comparing age-related differences between adult and pediatric populations?

2. Based on the statistical results of the length of stay parameter in table 4, the median and interquartile range (IQR)might be the more appropriate choices.

3. why so few of BDG test were performed in the current study

4. Why there are so many letter “a” in table 3 ?

5. �Page 10 ,line 1st � Please describe the diagnostic criteria for fungal infections in detail within the text, rather than simply citing references.

**Do you want your identity to be public for this peer review?** For information about this choice, including consent withdrawal, please see our Privacy Policy

Reviewer #1: No

Reviewer #3: No

Reviewer #4: No

---

## [Author Response · Author response to Decision Letter 2]

10 Jul 2025

Dear Dr. Nasir,

Thank you for accepting to review once again our manuscript entitled:

"A prospective, multicenter study of invasive fungal disease caused by molds in children and adults in Chile."

We greatly appreciate the insightful comments and suggestions provided by the four reviewers, which have helped us improve the quality of our manuscript.

Below, we provide our detailed responses to the reviewers’ major comments:

Major Comments

1. Missing Methods

1.1 Fungal culture conditions

Reviewer comment: The conditions under which “fungal culture” was performed should be clearly described.

Response: We have added more detail to the Methods section: “Fungal cultures were incubated at 30 °C on plates containing 2% Sabouraud dextrose agar and potato dextrose agar (Merck®, Germany). Incubation was maintained for up to 30 days, with daily macroscopic evaluations to monitor fungal growth and morphological characteristics.” (Page 8)

1.2 Source of lactophenol blue stain

Reviewer comment: The source (commercial or in-house) for the “lactophenol blue stain” should be specified.

Response: We clarified that the stain was commercially sourced: “Microscopic observation was performed using commercial lactophenol cotton blue stain (Merck®, Germany), applied between a slide and coverslip, and examined under light microscopy at 40X magnification.” (Page 8)

1.3 Tube protein extraction method

Reviewer comment: More detailed information should be provided regarding the “tube protein extraction method.”

Response: We included the following clarification: “Protein extraction was performed following the Bruker protocol for filamentous fungi, with minor modifications.” (Page 8)

2. Statistical comparison of disease frequencies

Reviewer comment: Consider conducting hypergeometric tests comparing disease frequencies to hospital admission data.

Response: We fully agree that this analysis would add value. However, data on the total number of patients with each condition (e.g., diabetes, leukemia, transplantation, etc.) at each participating hospital were not collected as part of the study design and are currently unavailable. We have acknowledged this limitation in the Discussion section and will consider including such data in future epidemiological studies. (Page 24)

3. Discussion

3.1 Mechanistic explanation for fusariosis and diabetes

Reviewer comment: Provide a biological or pathophysiological explanation for the observed associations.

Response: We expanded the discussion to include epidemiological associations between fusariosis and neutropenia, and between mucormycosis and poorly controlled diabetes. We also added a new reference to support the association of fusariosis in pediatric patients. (Pages 19–20)

3.2 Statistical support for viral pneumonia frequency

Reviewer comment: Consider statistical testing to assess the significance of viral pneumonia frequency.

Response: As mentioned above, we lack comprehensive data on viral pneumonia cases across all centers. We have acknowledged this limitation in the Discussion and emphasized the need to address it in future studies. (Page 24)

Reviewer #4 Comments

1. Clinical implications of age-related differences

Response: We aimed to explore differences between pediatric and adult populations and identified variations in associated conditions, fungal species, and survival rates. These findings support the need for age-specific diagnostic and therapeutic approaches.

2. Length of stay – statistical reporting

Response: We have updated Table 4 to include the median and interquartile range (IQR) for the length of stay variable.

3. Limited use of BDG testing

Response: BDG testing was not part of the standard diagnostic protocol and was only performed upon specific request by the treating team. We clarified this in the Methods (Page 9) and Results sections (Page 14).

4. Table 3 formatting – letter “a”

Response: We reviewed Table 3 and removed any unintended symbols or formatting artifacts that may have appeared as the letter “a.”

5. Diagnostic criteria description

Response: We expanded the description of the diagnostic criteria for fungal infections in the Methods section, rather than relying solely on references. (Page 10)

We hope that the revised manuscript and our responses adequately address the reviewers’ concerns. We sincerely thank you and the reviewers for your valuable feedback and consideration.

Sincerely,

Dr. Ricardo Rabagliati

Corresponding Author

---

## [Editor Report · Decision Letter 2]

1 Aug 2025

A prospective, multicenter study of invasive fungal disease caused by molds in children and adults in Chile

PONE-D-24-19295R2

Dear Dr. Rabagliati,

We’re pleased to inform you that your manuscript has been judged scientifically suitable for publication and will be formally accepted for publication once it meets all outstanding technical requirements.

Kind regards,

Nosheen Nasir

Academic Editor

PLOS ONE
---

## [Editor Report · Acceptance letter]

PONE-D-24-19295R2

PLOS ONE

Dear Dr. Rabagliati,

I'm pleased to inform you that your manuscript has been deemed suitable for publication in PLOS ONE. Congratulations! Your manuscript is now being handed over to our production team.

Kind regards,

on behalf of

Dr. Nosheen Nasir

Academic Editor

PLOS ONE